# Fever as the Only First Sign of Crohn’s Disease—Difficulties in Diagnosis during the COVID-19 Pandemic

**DOI:** 10.3390/children9121791

**Published:** 2022-11-22

**Authors:** Aleksandra Kęsicka, Judyta Burandt, Adam Główczewski, Aneta Krogulska

**Affiliations:** 1Student Research Club Paediatric, Allergology and Gastroenterology, Ludwik Rydygier Collegium Medicum in Bydgoszcz, Nicolaus Copernicus University in Toruń, 87-100 Toruń, Poland; 2Department of Pediatrics, Allergology and Gastroenterology, Collegium Medicum in Bydgoszcz, Nicolaus Copernicus University, 87-100 Toruń, Poland

**Keywords:** fever, Crohn’s disease, teleconsultations

## Abstract

We present a case of a nine-year-old girl with Crohn’s disease whose only first manifestation was fever. The patient was treated with antibiotics for six weeks by her general practitioner via teleconsultations during the COVID-19 pandemic. However, no significant improvement was observed. Only the appearance of loose stools after six weeks of observation and the lack of effect of previous treatment allowed for targeting of the diagnostic process and an unequivocal recognition of Crohn’s disease. Our aim is to emphasize the difficulties in diagnosis related to the atypical course of the disease, especially in the context of the ongoing COVID-19 pandemic. The described course of Crohn’s disease occurs in a minority of patients; however, this disease should not be overlooked in the differential diagnosis of fever in paediatric patients.

## 1. Introduction

Crohn’s disease (CD) is a type of inflammatory bowel disease (IBD) characterized by transmural inflammation of the bowel; however, it may also occur at any point in the gastrointestinal tract. Up to 30% of IBD patients are diagnosed in childhood or adolescence, particularly around seven years of age [1]. The median time for establishing a diagnosis of CD is 4.9 months [IQR 2.3–10.8 months] [2]. The most common symptoms of CD, which include diarrhea, abdominal pain, rectal bleeding, fever, weight loss and fatigue, are not highly specific when considered individually [3]. Fever is the main symptom that occurs in countless diseases (Table 1) [4]. As the only initial symptom of IBD, it occurs in 10–15% of patients [5]. Non-specific clinical presentation may delay diagnosis, especially during pandemic, when fever in children was often associated with COVID-19 [6]. The diagnosis of IBD was also significantly constrained during the COVID-19 pandemic because traditional visits to the GP were replaced by telephone consultations.

## 2. Case Presentation

A 9-year-old girl, without any significant medical history, was admitted to the Department of Paediatrics, Allergology and Gastroenterology with a fever up to 39 °C, alternating with subfebrile temperature, which lasted for about six weeks. No other symptoms were reported except for poor appetite persisting from early childhood. She passed a stool of the right consistency, without pathological impurities, once a day. When the patient developed a fever, the general practitioner (GP) suspected pharyngitis and ordered antibiotic therapy during a teleconsultation. No improvement was observed, and the antibiotic therapy was modified several times in the following teleconsultations, without any significant improvement (Figure 1). Loose stools without pathological impurities later appeared. After excluding COVID-19, the patient was admitted to the Department of Paediatrics, Allergology and Gastroenterology. On the next day, the girl started to pass one to two stools daily (Bristol type 2), without blood.

Upon physical examination, the patient’s general condition was assessed to be quite good aside from being underweight (BMI < 3 c; height = 25 c; weight < 3 c) with pale skin. Laboratory tests revealed anaemia, elevated levels of inflammatory markers, calprotectin, RF, ASCA and FOBT (Table 2). Infectious gastrointestinal tract diseases, including noroviruses, adenoviruses, rotaviruses, Candida spp., Clostridioides difficile and Campylobacter jejuni, were comprehensively excluded. Blood and urine cultures were negative.

After excluding other causes of fever, it was decided to focus on digestive system diseases.

Abdominal ultrasound examination showed a thickening of the terminal ileum up to 5 mm, a thickening of the Bauhin valve wall up to 4 mm and hyperaemia. Computed tomography of the abdominal cavity revealed a slight enlargement of the ileocecal lymph nodes. Gastroscopy revealed signs of inflammation along the entire length of the oesophagus, with the presence of fine erosion plaque and whitish, fine deposits above the cardia. Colonoscopy revealed a swollen ileocecal valve, intense inflammatory changes in the terminal ileum and numerous ulcerations with non-specific morphology in the ascending colon and transverse colon. Histopathological examination confirmed a small number of lymphocytic infiltrates in the mucosa. No typical morphological features of specific and non-specific inflammatory bowel disease or CMV infection were observed.

Until that point, the patient had received empirical broad-spectrum antibiotic therapy; however, after eight weeks of antibiotic therapy, based on the test results and an observed lack of improvement, systemic steroid therapy with methylprednisolone (1 mg/kg/day i.v.) was commenced, resulting in the fever subsiding and inflammatory marker levels falling. Considering the entire clinical picture, Crohn’s disease (CD) was suspected. The patient was discharged home with the recommendation of continuing oral steroid therapy with prednisone (1.25 mg/kg/day p.o.).

One week after discharge, the patient was re-admitted to the clinic due to a recurrence of high fever, poor response to antipyretic drugs and loose stools (3 per day, without blood). Physical examination revealed the same abnormalities as during the previous hospitalization. Magnetic resonance imaging with enterography showed active inflammatory lesions in the distal ileum. Due to the persistent fever and elevated CRP, it was decided to intensify the antibiotic therapy by adding piperacillin with tazobactam, and the oral steroids were replaced with the intravenous form. The fever subsided two days after modifying the antibiotic and steroid therapies, and a gradual reduction of inflammatory markers was observed. After 10 days, the antibiotic therapy was discontinued.

A repeated gastroscopic examination revealed a normal picture, without the previously described changes. The colonoscopy revealed oedema of the ileocecal valve and the terminal ileum, reddening of the mucosa and multiple, deep, irregular ulcers with fibrin-covered bases. The observed inflammatory features extended from the lower part of the cecum to the hepatic flexure of the colon. Single inflammatory polyps were also present. Histopathological examination revealed visible ulcerations in the mucosa of the ileum and reactive hyperplasia of the lymphoid system; the mucosa of the large intestine demonstrated features of slight oedema, with diffuse infiltration of lymphocytes, plasmocytes and eosinophils in the lamina propria. One specimen also showed numerous neutrophil infiltrates, however, no granuloma was observed. The results of the endoscopic and pathomorphological examinations allowed an unequivocal diagnosis of Crohn’s disease.

On the third day after the cessation of antibiotic therapy, the patient reported another fever and continued to pass loose stools (3 per day, without blood). Due to lack of improvement, biological treatment with adalimumab was implemented. After 24 days of treatment, the patient was in good general condition, without any complaints, and was discharged home.

## 3. Discussion

Fever is a particularly common presentation in a wide spectrum of infectious and non-infectious diseases. In most cases, acute fever is caused by a mildly serious bacterial infection or a self-limiting viral infection and resolves after about a week. However, if this is not the case, it may indicate many other causes which should be considered during the diagnosis process. During the COVID-19 pandemic, the main cause of fever was often suspected to be SARS-CoV-2 infection; indeed, fever was reported to be the only symptom in 47.5% of paediatric patients with COVID-19 [7].

In the described case, the patient was treated for persistent fever with antibiotics which were modified several times for about six weeks. Due to the epidemiological situation in the country, these modifications occurred via teleconsultations. The unquestionable disadvantage of this form of communication with patients is the inability to perform a physical examination, which could affect the diagnostic process [8]. It is possible that a physical examination by a GP could help establish a diagnosis more quickly.

Based on medical history, physical examination, abnormalities in lab tests, endoscopy results and histopathological examination it was possible to establish the final diagnosis. It is noteworthy that the patient did not experience symptoms typical of CD over a long period. In some cases, the appearance of fever may precede the appearance of typical IBD symptoms by many months [4].

Fever as the only initial symptom of CD occurs in a minority of patients [5]. The described case shows an atypical clinical manifestation of the onset of Crohn’s disease which coincided with the COVID-19 pandemic. It is significant to note that even though lockdowns have been lifted, many clinicians are still doing remote consultations for patient comfort, so this is still an important consideration. Based on the presented case it can be concluded that if the fever persists after the first course of antibiotic therapy, it is important to conduct a face-to-face consultation that includes a physical examination. The case emphasizes that even during the pandemic, any child presenting with persistent fever, especially without any other apparent signs of infection, should be managed carefully and that practitioners should consider less common causes, including CD.

## Figures and Tables

**Figure 1 children-09-01791-f001:**
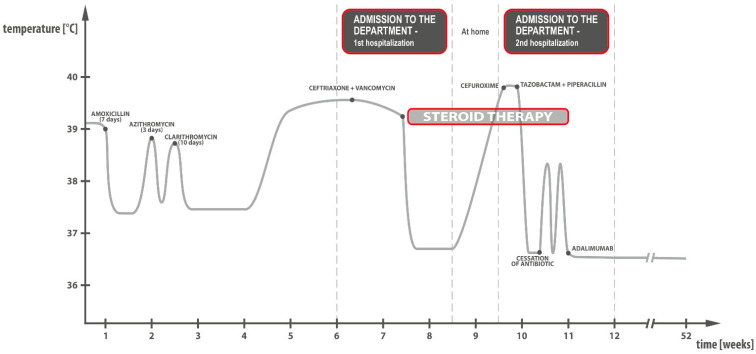
Course of the fever and treatment administered to the patient.

**Table 1 children-09-01791-t001:** Causes of recurrent fever in pediatric patients [4].

Infectious Causes	Noninfectious Causes
**Viral diseases:** Repeated independent respiratory viral infections; Parvovirus B19 infection; Epstein-Barr (EBV) virus infection; Recurrent herpes virus infection.	**Immune-mediated and granulomatous diseases:** Crohn’s disease; Behҫet disease; Systemic lupus erythematosus (SLE); Juvenile dermatomyositis (JDM); Acute rheumatic fever; Leukoclastic angiitis syndromes; Sarcoidosis; Granulomatous hepatitis.
**Bacterial diseases:** Relapsing fever (Borrelia recurrentis and other borreliae); Brucellosis; Trench fever (Bartonella quintana); Syphillis (Treponema pallidum); Rat bite fever (Spirillum minus); Melioidosis (Burkholderiapseudomallei); Whipple disease; Chronic meningococcemia; Infective endocarditis; Subacute cholangitis; Abscesses, especially occult dental abscesses; Osteomyelitis; Tuberculosis.	**Periodic fever syndromesand autoinflammatory disorders:** Cyclic neutropenia; Periodic fever, aphthous stomatitis, pharyngitis, adenopathy (PFAPA) syndrome; Familial Mediterranean fever (FMF); Hyperimmunoglobulinemia D with periodic fever syndrome (HIDS); Cryopyrin-associated periodic syndromes (CAPS); Familial cold autoinflammatory syndrome (FCAS); Muckle-Wells syndrome (MWS); Neonatal onset multisystem inflammatory disease (NOMID); TNF receptor-associated periodic syndrome (TRAPS); Systemic juvenile idiopathic arthritis (sJIA).
**Fungal diseases:** Histoplasmosis; Coccidioidomycosis.	**Neoplasms**
**Parasitic diseases:** Malaria; Visceral leishmaniasis.	**Hypersensitivity diseases:** Hypersensitivity pneumonitis; Drug fever; Weber-Christian disease (panniculitis).
	**Other conditions:** Sweet syndrome; Fabry disease; Congenital insensitivity to pain with anhidrosis; Anhidrotic ectodermal dysplasia; Sickle cell crisis; Castleman disease; Erdheim-Chester disease; Kikuchi-Fujimoto disease; Diabetes insipidus; Central nervous system abnormalities; Factitious fever.

**Table 2 children-09-01791-t002:** Laboratory test results.

Laboratory Test	Beggining of the 1st Hospitalization	Beggining of the 2nd Hospitalization	After Starting Biological Treatment
**Complete blood count:**			
WBC [10^3^/µL]	6.84	7.85	4.69
RBC [10^6^/µL]	4.51	3.85	4.98
HGB [g/dL]	9.00 ↓	8.00 ↓	10.3 ↓
HCT [%]	31.1 ↓	27.6 ↓	34.5
MCH [pg]	20.0 ↓	20.8 ↓	20.7 ↓
MCV [fl]	69.0 ↓	71.7 ↓	69.3 ↓
MCHC [g/dL]	28.9 ↓	29.0 ↓	29.9 ↓
PLT [10^3^/µL]	482 ↑	357 ↑	358 ↑
NEUTROPHILS [10^3^/µL]	4.37	4.82	1.39 ↓
LYMPHOCYTES [10^3^/µL]	1.0	1.53	2.51
MONOCYTES [10^3^/µL]	1.31 ↑	1.18 ↑	0.49
EOSINOPHILS [10^3^/µL]	0.11	0.28	0.25
BASOPHILS [10^3^/µL]	0.05	0.04	0.05
CRP [ng/mL]	278.1 ↑	146.26 ↑	0.88
PCT [ng/mL]	1.70 ↑	0.59 ↑	
ALT [U/l]	7	13	19
AST [U/l]	12	15	29
ANCA	-		
ASCA	+		
RF	+		
Anti-tTG IgA antibodies	-		
SARS-CoV-2	-	-	-
FOBT	+	+	-
Faecal calprotectin [mg/kg]	1514.5 ↑	506.8 ↑	

WBC—white blood cells, RBC—red blood cells, HGB—hemoglobin, HCT—hematocrit, MCH—mean corpuscular hemoglobin, MCV—mean corpuscular volume, MCHC—mean corpuscular hemoglobin concentration, PLT—platelet count, CRP—C-reactive protein, PCT—procalcitonin, ALT—alanine aminotransferase, AST—aspartate aminotransferase, ANCA—anti-neutrophil cytoplasmic antibodies, ASCA—anti-Saccharomyces cerevisiae antibodies, RF—rheumatoid factor, FOBT—fecal occult blood test, ↑—elevated level, ↓—decreased level.

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
