# Peer review of "Fever as the Only First Sign of Crohn’s Disease—Difficulties in Diagnosis during the COVID-19 Pandemic"

_children, 2022, doi:10.3390/children9121791_

Round 1
Reviewer 1 Report
Nice reminder of fever as main presentation in patient with Crohn's disease.
I particularly like your graph showing how fever settled with treatment.
Did the patient explicitly deny diarrhea in initial phone consultations ie did the doctor ask if patient had diarrhea?
Please add a little more detail about type and dose of steroids used.
Please state if granuloma seen on histology or not
Please mention why EEN was not considered as a treatment
At presentation to the hospital you mention patient had lost of appetite and loose stool- please give more detail eg how much weight lost and how frequent and what type of stool . Was this initially thought to be related to antibiotic side effect?
Conclusion:
please avoid just repeating the case history- rewrite those sentences that just repeat the history
Please rephrase this sentence-
'should not rule out the possibility of less frequent etymologies, including CD'
Suggest
should consider alternative less common causes including CD
Also may want to add in that even though lock downs have been lifted lots of clinicians are still doing remote consultations for patient ease etc- so this is still any important consideration.
Perhaps make a recommendation that if fever not settling as expected after first course of antibiotic face to face review should take place to allow examination
Reviewer 2 Report
The authors of this study try to present a rare and unique case of Crohn's Disease in a child with fever as the only first manifestation. I think the topic of this study is interesting and can add more evidences into the existing literature. However, there are some issue within the manuscript which warrant further clarifications and revisions. Below you can see my detailed comments of this manuscript:
1) This patient was presented with fever at first and went to the general practitioner where she received several antibiotics treatment. After the antibiotic treatments, the symptoms of loose stool appeared. Inflammatory Bowel Disease (IBD) itself can also be caused or exacerbated by medications, such as antibiotics. Can the Crohn's Disease in this patient, by any chance, caused by antibiotics administration? Can this patient experience both Crohn's Disease and infections at the first time?
2) Did this patient undergo RT-PCR test when she had fever or during hospital admission? If yes, the results should be provided. If no, why?
3) "Until that point, the patient had received empirical broad-spectrum antibiotic therapy; however, after eight weeks of antibiotic therapy." Why did this patient still receive broad-spectrum antibiotic therapy if there were no signs of infection, instead laboratory, radiological, and histopathological results support the diagnosis of Crohn's Disease? Why must wait until eight weeks course of antibiotics treatment before starting steroid treatment?
4) "Based on the entire clinical picture, Crohn’s disease (CD) was suspected." Given the results of laboratory, radiological, colonoscopy, and histopathological examinations from this patient, why was the diagnosis of Crohn's disease (CD) still suspected and couldn't be established yet? What further examinations needed by the authors to make a definite diagnosis of Crohn's Disease?
5) Is there any follow-up? How's the condition of the patient during the follow-up period? Do the symptoms recur?
